# Functional Characterization of the *N*-Acetylmuramyl-l-Alanine Amidase, Ami1, from *Mycobacterium abscessus*

**DOI:** 10.3390/cells9112410

**Published:** 2020-11-04

**Authors:** Tanja Küssau, Niël Van Wyk, Matt D. Johansen, Husam M. A. B. Alsarraf, Aymeric Neyret, Claire Hamela, Kasper K. Sørensen, Mikkel B. Thygesen, Claire Beauvineau, Laurent Kremer, Mickaël Blaise

**Affiliations:** 1Institut de Recherche en Infectiologie de Montpellier (IRIM), Université de Montpellier, CNRS UMR 9004, CEDEX 5, 34293 Montpellier, France; tanja.kuessau@gmail.com (T.K.); nielvw@gmail.com (N.V.W.); Matt.Johansen@uts.edu.au (M.D.J.); husam.alsarraf@irim.cnrs.fr (H.M.A.B.A.); claire.hamela@irim.cnrs.fr (C.H.); 2Department of Molecular Biology and Genetics, Aarhus University, 8000 Aarhus C, Denmark; 3CEMIPAI CNRS UM UMS3725, CEDEX 5, 34293 Montpellier, France; aymeric.neyret@irim.cnrs.fr; 4Department of Chemistry, Faculty of Science, University of Copenhagen, Thorvaldsensvej 40, DK-1871 Frederiksberg C, Denmark; kso@chem.ku.dk (K.K.S.); mbth@chem.ku.dk (M.B.T.); 5Chemical Library Institut Curie/CNRS, CNRS UMR9187, INSERM U1196 and CNRS UMR3666, INSERM U1193, Université Paris-Saclay, F-91405 Orsay, France; Claire.Beauvineau@curie.fr; 6INSERM, IRIM, 34293 Montpellier, France

**Keywords:** *N*-acetylmuramyl-l-alanine amidase, peptidoglycan, drug screening, x-ray crystallography, *Mycobacterium abscessus*

## Abstract

Peptidoglycan (PG) is made of a polymer of disaccharides organized as a three-dimensional mesh-like network connected together by peptidic cross-links. PG is a dynamic structure that is essential for resistance to environmental stressors. Remodeling of PG occurs throughout the bacterial life cycle, particularly during bacterial division and separation into daughter cells. Numerous autolysins with various substrate specificities participate in PG remodeling. Expression of these enzymes must be tightly regulated, as an excess of hydrolytic activity can be detrimental for the bacteria. In non-tuberculous mycobacteria such as *Mycobacterium abscessus*, the function of PG-modifying enzymes has been poorly investigated. In this study, we characterized the function of the PG amidase, Ami1 from *M. abscessus*. An *ami1* deletion mutant was generated and the phenotypes of the mutant were evaluated with respect to susceptibility to antibiotics and virulence in human macrophages and zebrafish. The capacity of purified Ami1 to hydrolyze muramyl-dipeptide was demonstrated in vitro. In addition, the screening of a 9200 compounds library led to the selection of three compounds inhibiting Ami1 in vitro. We also report the structural characterization of Ami1 which, combined with in silico docking studies, allows us to propose a mode of action for these inhibitors.

## 1. Introduction

Peptidoglycan (PG), composed of glycan chains cross-linked by short peptides, surrounds most bacteria and protects them against environmental and osmotic stress. PG consists of polymers of the *N*-acetylglucosamine (GlcNAc) and *N*-acetylmuramic acid (MurNAc) disaccharide units. The three-dimensional PG network is formed by the cross-linking of the peptide stems attached to the MurNAc moiety [1]. Although contributing to the rigidity of the cell envelope, PG remains a highly-dynamic structure participating in important physiological processes, such as bacterial elongation and separation during cell division [2]. During these dynamic phases, PG is remodeled and the balance between PG synthesis and hydrolysis is tightly regulated [3]. PG remodeling is performed by numerous autolysins acting on specific substrates and at different stages during bacterial growth. While glycosidases, such as muraminidases and glucosaminidases act to hydrolyze the glycosidic bonds between the carbohydrate units, peptidases, such as *N*-acetylmuramyl-l-alanine amidases, l,d-endopeptidases, d,l-endopeptidases, l,d-carboxypeptidases, and dd-carboxypeptidases, hydrolyze the amide bonds of the peptide stem at specific positions [4]. 

In mycobacteria, PG differs from Gram-positive or Gram-negative bacteria as it is covalently linked to the atypical mycobacterial envelope [5]. d,l-endopeptidases belonging to the papain-like peptidase superfamily, possessing an NlpC/P60 catalytic domain [6], have been widely studied in mycobacteria. Among them, RipA has been shown to interact with the resuscitation-promoting factor B (RpfB), a lytic transglycosylase, and the bifunctional PG-synthesizing enzyme, penicillin-binding protein 1 (PBP1) [7]. Rpf-interacting protein A (RipA) activation also requires the *Mycobacterium* acid resistance protease (MarP) during acid stress [8]. These studies emphasize the coordination of the PG hydrolytic enzymes, which are tightly regulated during the mycobacterial life cycle. 

Until recently the role of the *N*-acetylmuramyl-l-alanine amidases (amidases) has been poorly documented in mycobacteria. The genomes of *Mycobacterium smegmatis* and *Mycobacterium tuberculosis* encode four potential amidases. CwlM (Rv3915), also known as Ami2, is annotated as a PG amidase but does not appear to be catalytically active [9]. However, it plays an important role in modulating the activity of the UDP-*N*-acetylglucosamine 1-carboxyvinyltransferase MurA, a key player of PG biosynthesis [9]. Rv3811/MSMEG_6406, designated Ami3, is not essential for mycobacterial growth, but overexpression of Ami3, due to its inherent PG hydrolytic activity, is lethal for mycobacteria [10]. Ami3 activity is therefore tightly modulated by proteolysis mediated by the HtrA protease [10]. MSMEG_6281 (Ami1_Msm_), containing an amidase_3 domain, is dispensable for bacterial growth but *ami1* deletion mutants present an atypical lateral branching and a defect in cell separation and septal PG turnover [11]. This mutant is also more susceptible to antibiotics due to increased cell wall permeability; a phenotype subsequently confirmed in an independent study which also reported the attenuation of the virulence the *ami1* mutant in a mouse model of infection [12]. While cell division was impaired in the *ami1_Msm_* knockout strain, this phenotype could be rescued by complementation with the orthologue gene, *Rv3717*, from *M. tuberculosis* (*ami1_Mtb_*), suggesting a similar mode of action in both species [13]. Furthermore, *ami1_Msm_* has also been shown to be important for biofilm formation [14]. In addition, mycobacterial Ami1 has been biochemically and structurally characterized. Purified Ami1_Mtb_ hydrolyzes PG fragments in vitro in a zinc-dependent manner and its crystal structure revealed a typical amidase_3 signature [15,16]. However, the functional role of Ami1 in mycobacteria appears more complex than originally thought, for several reasons: (1) a single *ami1* deletion mutant or multiple deletions of *ami1* and *ami3* or *ami4* do not display any growth defect in *M. smegmatis* [10], suggesting that amidases have redundant functions as reported in other bacteria, such as in *Escherichia coli* [17,18,19] and (2) Ami1 shares overlapping functions with the endopeptidase RipA, as Ami1 rescued RipA’s activity during cell division but not for persistence and chronic infection in a mice infection model [20]. 

The PG remodeling machinery of *M. abscessus,* a rapidly growing non-tuberculous mycobacterium, has been poorly explored. The *M. abscessus* complex [21] comprises three different subspecies—*M. abscessus sensu stricto* (hereafter designated *M. abscessus*)*, M. abscessus subsp. bolletii,* and M*. abscessus subsp. massiliense.* Infections caused by this complex are increasing globally and are notoriously difficult to treat, particularly due to the extreme resistance of *M. abscessus* to most antibiotic classes [21]. The two *M. abscessus* morphotypes, (R) rough and (S) smooth, display similar antibiotic susceptibility profiles but, largely differ in their virulence profile as emphasized in the embryonic zebrafish model [22]. The S variant is considered as the environmental form that colonizes the airways and that can transition to the more virulent R form [21]. The presence or absence of surface-exposed glycopeptidolipids (GPL) in both the S and R morphotypes, respectively, has been shown to correlate with virulence in several animal models as well as in humans [23,24,25].

Herein, we evaluated the impact of *ami1_Mab_* deletion with respect to growth and bacterial division in both S and R morphotypes of *M. abscessus* and assayed their capacity to infect human macrophages and zebrafish larvae. We characterized the hydrolytic properties of purified Ami1_Mab_ and determined its crystal structure at high resolution. We also report a thermal-shift assay screening approach that led to the discovery of three Ami1_Mab_ inhibitors. 

## 2. Materials and Methods

### 2.1. Generation of Unmarked Deletion Mutant

Deletion of *MAB_0318c* (*ami1_Mab_*) in the wild-type strain *M. abscessus* CIP104536^T^ was carried out using the double homologous recombination unmarked strategy, reported previously [26]. The DNA sequences located upstream, left arm (LA), and downstream, right arm (RA), of the *ami1* gene were PCR-amplified and subsequently cloned after an overlapping PCR reaction into the pUX1*-katG* using the following primers, Forward: Mab0318_pUX1katG_LA_F and Reverse: Mab0318_pUX1katG_LA_R and Forward: Mab0318_pUX1katG_RA_F and Reverse: Mab0318_pUX1katG_RA_R (Appendix A).

The resulting plasmid was electroporated in the smooth (S) and rough (R) variants of *M. abscessus* and the red fluorescent, kanamycin (KAN)-resistant colonies which underwent the first homologous recombination event were selected on Middlebrook 7H10 (Sigma-Aldrich, Saint-Quentin Fallavier, France) agar supplemented with 10% oleic acid-albumin-dextrose-catalase (OADC) in the presence of 200 μg/mL KAN. Cultures of selected colonies were then subjected to a counter-selection on 7H10 agar plates supplemented with OADC containing 50 μg/mL isoniazid (INH) to promote a second homologous recombination and removal of the KAN cassette. Non-fluorescent, INH-resistant colonies were picked up and further genotyped for *ami1_Mab_* deletion following PCR using the Mab0318_pUX1katG_LA_F and Mab0318_pUX1katG_RA_R primers (Appendix A). Proper deletion was further confirmed by DNA sequencing of the genomic region surrounding the site of gene deletion. 

For functional complementation of the *ami1_Mab_* deletion mutant, the genomic sequence of *ami1_Mab_* was fused at its 3′end in frame with the sequence encoding the Streptag-II, using the following primers, Forward: Mab0318_pMV306hsp_EcoRI_F and Reverse: Mab0318_SalI_C-termStrepTag (Appendix A) and subsequently cloned into the pMV306 plasmid possessing a KAN-resistance cassette and resulting in the creation of pMV306::*ami1_Mab_*. The *ami1_Mab_* deletion mutants were transformed with pMV306::*ami1_Mab_* and selected onto 7H10 agar supplemented with OADC and KAN (200μg/mL). 

### 2.2. Cloning for Recombinant Protein Expression

The genomic sequence encoding *ami1_Mab_* depleted from its secretion peptide signal was PCR-amplified from *M. abscessus* genomic DNA with the following primers, Forward: Mab0318_pET151_F and Reverse: Mab0318_pET151_R (Appendix A) and cloned in a ligation-free independent manner into the directional Champion™ pET151 TOPO™ (Thermo Fisher Scientific, Villebon-sur-Yvette, France) using manufacturer’s instructions. 

The genomic sequence of *ami1_Msm_* without its peptide signal of secretion was PCR-amplified from *M. smegmatis* mc^2^155 genomic DNA with the Forward and Reverse primers, Smeg6281_pET30_F and Smeg6281_pET30_R (Appendix A) and was cloned into the pET-30 ek/LIC plasmid between the KpnI and EcoRI restriction sites in frame with the His- and S-tags followed by a Tobacco etch virus (TEV) protease cleavage site. 

The genomic sequence of *ami1_Mtb_* without its peptide signal of secretion was PCR-amplified from *M. tuberculosis* genomic DNA with the Forward Mtb_3717_F and Reverse Mtb_3717_R primers (Appendix A) and was cloned in a ligation-free independent manner into the pET-30 ek/LIC plasmid in frame with the His- and S-tags followed by a Tobacco etch virus (TEV) protease cleavage site. 

### 2.3. Mycobacterial Culture Conditions

The MIC were determined according to the Clinical & Laboratory Standards Institute (CLSI) guidelines [27]*. M. abscessus* strains were grown in Middlebrook 7H9 broth (BD Difco) supplemented with 10% OADC and 0.025% tyloxapol (Sigma-Aldrich, Saint-Quentin Fallavier, France) at 37 °C under agitation (unless otherwise stated) in the presence of antibiotics, when required (750 µg/mL hygromycin for all tdTomato-expressing strains, and 200 µg/mL kanamycin for complemented strains). On plates, strains were grown on Middlebrook 7H10 agar (BD Difco) supplemented with 10% OADC enrichment in the presence of antibiotics when required and grown at 37 °C.

### 2.4. Single-Cell Preparation for M. abscessus 

The various strains expressing tdTomato were grown in 50 mL 7H9 supplemented with 10% OADC, 0.025% tyloxapol and appropriate antibiotics (750 µg/mL hygromycin and 200 µg/mL KAN for complemented strains) for 3 days at 37 °C under agitation. After bacteria were centrifuged for 15 min at 3000× *g*, the pellet was resuspended in 1 mL of 7H9 and transferred into 1.5 mL Eppendorf tubes. To dissociate bacterial aggregates and generate single-cell bacteria, the samples were syringed through a 26G needle 20 times followed by a 15 s sonication step, resuspended with 7H9 media and passed through a 5 µm filter. Bacterial suspensions were centrifuged for 15 min at 3000× *g* and the pellet resuspended in 1 mL of 7H. The OD was measured and 10 µL aliquots were prepared in 1.5 mL Eppendorf tubes and stored at −80 °C.

### 2.5. Protein Expression and Purification 

The *E. coli* BL21(DE3) strain resistant to phage T1 (New England Biolabs, Evry, France) containing the pRARE2 plasmid was transformed with the pET-151::*ami1_Mab_*, pET-30::*ami1_Msm_*, or pET-30::*ami1_Mtb_* constructs. Bacterial cultures were performed in Luria-Bertani broth (LB) medium containing 200 μg/mL ampicillin or 50 μg/mL KAN and 30 μg/mL chloramphenicol at 37 °C. When an OD at 600 nm of 0.6 to 1 was reached, the cultures underwent a 30 min cold shock in icy water before adding 1 mM isopropyl-β-d-1-thiogalactopyranoside (IPTG). Cultures were then grown for 20 h at 16 °C and then spun-down (8400× *g*, 4 °C, 20 min). Bacterial pellets were resuspended in lysis buffer (50 mM Tris-HCl pH 8, 400 mM NaCl, 20 mM imidazole, 1 mM benzamidine, and 5 mM β-mercaptoethanol) and the cells were disrupted by sonication prior to clarification by an additional centrifugation step (27,000× *g*, 4 °C, 45 min). Proteins were then purified by a first ion metal affinity chromatography (IMAC) step (Ni-nitrilotriacetic acid sepharose, Cytiva, Europe, GmbH). To remove the *N*-terminal tags, the eluted proteins were mixed in a 1:50 (*w:w*) ratio with TEV protease and dialyzed overnight at 4 °C against 50 mM Tris-HCl pH 8, 200 mM NaCl, and 5 mM β-mercaptoethanol. The dialyzed proteins were loaded onto a second IMAC, the non-tagged proteins were eluted in the flow-through, while tags, uncleaved protein, and TEV protease bound the column. Proteins were concentrated using a 10 kDa cut-off centricon (Sartorius) and subjected to a final step of purification on size-exclusion chromatography, Supderdex 75 10/300 GL using an elution buffer containing 50 mM Tris-HCl pH 8, 200 mM NaCl, and 5 mM β-mercaptoethanol for Ami1_Mab_ and Ami1_Mtb_ or the same buffer but at pH 7.4 for Ami1_Msm_. The proteins prepared in these conditions were for crystallization purpose. Ami1_Mab_ and Ami1_Mtb_ used for the thin-layer chromatography (TLC)-based activity assay were purified under similar conditions, except the last step of purification on SEC was performed in 1 x Phosphate-Buffered-Saline (PBS) buffer. 

### 2.6. Crystallization, Data Collection, and Structure Refinement 

Crystals were grown in sitting drops in MR Crystallization Plates™ (Hampton Research) at 18 °C by mixing 2 µL of protein solution with 2 µL of reservoir solution. Ami1_Mab_ crystals were obtained at a protein concentration of 10 mg/mL and crystallized in 0.1 M Bis-Tris pH 5.5, 1.6 M ammonium sulfate, and 1% (*w*/*v*) polyethylene glycol (PEG). Ami1_Mab_ plus dipeptide crystals were obtained in similar conditions and in presence of 5 mM of the ligand. Crystals were grown for several months and cryo-protected by a brief soaking step into a solution made of 0.1 M Bis-Tris pH 5.5, 1.6 M ammonium sulfate, 1% (*w*/*v*) PEG 3350, and 10–15% glycerol prior being cryo-cooled in liquid nitrogen.

Ami1_Msm_ crystals were obtained at a concentration of 13 mg/mL in the presence of 5 mM muramyl-dipeptide (MDP) and in a reservoir solution made of 0.1 M Bis-Tris pH 5.5, and 17% (*w*/*v*) PEG. Crystals were cryo-protected by a brief soaking step into a solution made of 0.1 M Bis-Tris pH 5.5, 30% (*w*/*v*) PEG 3350, and 10% PEG 400 prior being cryo-cooled in liquid nitrogen.

X-ray diffraction data were collected at the Swiss-light source on the X06DA-PXIII beamline and at the European Synchrotron Radiation Facility on the ID30B beamline. 

The Ami1_Mab_ structure was solved by molecular replacement using the crystal structure of the Ami1_Mtb_ structure (PDB: 4LQ6) as a search request. The Ami1_Mab_ dipeptide and Ami1_Msm_ structures were solved similarly but using the Ami1_Mab_ structure as a search model. Autobuild from the Phenix (version 1.18.2) software suite [28] was used to automatically improve the models which were then manually rebuilt and refined using the Coot (version 0.89) [29] and Phenix software suites. The coordinates and structures factors have been deposited at the protein data bank under accession numbers 7AGL for Ami1_Mab_, 7AGO for Ami1_Mab_ plus dipeptide, and 7AGM for Ami1_Msm_.

### 2.7. In Silico Docking 

The three-dimensional structure of Ami1_Mab_ bound to dipeptide was depleted from its ligand and was used for in silico docking. The docking search was performed with the PyRx software (version 0.9.4) [30] running AutoDock Vina [31]. All the amino acid side chains were kept as rigid. Docking was carried out using the following grid parameters: center X = 0.59, Y = 16.38, and Z = −8.5 and dimensions (Å) X = 40, Y = 44, and Z =28 and an exhaustiveness of 8. 

### 2.8. Drug Susceptibility Testing and MIC Determination 

Minimum inhibitory concentrations (MICs) were determined in cation-adjusted Mueller–Hinton broth (Sigma-Aldrich, Saint Quentin Fallavier, France) using the microdilution method. Briefly, 2 μL of the drug was added to 198 μL bacterial suspension and 2 times serial dilutions were performed with a final volume of 100 μL per well (5 × 10^6^ colony forming unit CFU/mL). The MIC was defined as the minimal drug concentration required to inhibit growth and was recorded by visual inspection after 4 days of incubation at 30 °C. All experiments were completed in technical replicates and performed in three independent replicates.

### 2.9. Macrophage Experiments 

THP-1 macrophages were grown in Roswell Park Memorial Institute (RPMI) medium (BD Gibco) supplemented with 10% fetal bovine serum (FBS) (Sigma-Aldrich, Saint Quentin Fallavier, France) (RPMI^FBS^) and incubated at 37 °C with 5% CO_2_. THP-1 monocytes were differentiated into macrophages in the presence of 20 ng/mL phorbol myristate acetate (PMA) in 24-well flat-bottom tissue culture microplates (10^5^ cells/mL) and incubated for 48 h at 37 °C with 5% CO_2_. Macrophages were infected with *M. abscessus* strains at a multiplicity of infection (MOI) of 2:1 (bacteria:macrophage) in 500 µL of RPMI^FBS^ for 2–4 h at 37 °C with 5% CO_2_. Following infection, the bacterial inoculum was removed and cells were carefully washed three times with PBS and then incubated with fresh RPMI^FBS^ supplemented with 250 µg/mL amikacin for 2 h to kill extracellular bacteria. Macrophages were then washed three times with PBS prior to the addition of 500 µL RPMI^FBS^ supplemented with 50 µg/mL amikacin to prevent extracellular growth. At various time points, macrophages were washed three times with PBS and lysed with 100 µL of 1% Triton X-100. Serial dilutions were plated on LB agar to count the colony-forming units. 

### 2.10. Synthesis of Peptidoglycan Dipeptide, l-Alanyl-d-Isoglutamine (DP)

The dipeptide was prepared by Fmoc solid-phase peptide synthesis and was performed on a Biotage^®^ Initiator+ AlstraTM microwave-assisted, automated peptide synthesizer from Biotage AB, Uppsala, Sweden. Amino acids, reagents, and solvents were purchased from Iris Biotech GmbH, Marktredwitz, Germany. Peptide couplings were carried out using 5 equivalents of amino acid, 5 equivalents of 1-hydroxybenzotriazole (HOBt), 4.75 equivalents of *N-*[(1*H-*benzotriazol-1-yl)(dimethylamino)methylene]-*N*-methylmethanaminium hexafluorophosphate *N*-oxide (HBTU), and 9.75 equivalents of *N*,*N*-diisopropylethylamine (DIEA) in *N*-methylpyrrolidone (NMP). A coupling time of 10 min at 75 °C, and subsequent deprotection with piperidine in *N*,*N*-dimethylformamide (DMF) (2:3) for 3 min, followed by piperidine-DMF (1:4) for 10 min was used. After each coupling and deprotection step, a washing procedure with 2x NMP, 1x dichloromethane, then 2 x NMP was used. 

The synthesis was carried out on a 0.1 mmol scale on 0.25 mmol/g Tenta Gel S Rink resins from Rapp Polymere GmbH, Tuebingen, Germany. Peptide coupling of Fmoc-d-Glu(*t*Bu)-OH was followed by Fmoc-d-Ala-OH. The peptide was cleaved from the resin with trifluoroacetic acid–water–triethylsilane 95:3:2 for two hours. Because of the high polarity of the product, purification was performed in reversed-phase mode on an Isolera™ system (Biotage AB, Uppsala, Sweden) with a SNAP Bio C18 300 Å column (Biotage AB, Uppsala, Sweden), with water–acetonitrile containing 0.1% TFA, and using a gradient of acetonitrile from 5% to 30%. Peptide purity was evaluated by analytical HPLC (Dionex Ultimate 3000 system from Thermo Scientific, Waltham, MA, USA) on an analytical C18 column (Gemini NX 110 Å, 5 μm, 4.6 × 50 mm from Phenomenex, Torrance, CA, USA) using a linear gradient flow from 2% to 25% acetonitrile, containing 0.1% formic acid, giving a purity >98%, and identification was carried out by electrospray ionization mass spectrometry using an Impact HD UHR-QTOF mass spectrometer from Bruker Corp., Billerica, MA, USA. MS (ESI): calculated for C_8_H_15_N_3_O_4_: 217.1063; found: [M+H]^+^ 218.1132, and [M+Na]^+^ 240.0956.

### 2.11. Chemical Library

The Institut Curie-CNRS library (9200 compounds, 2013 version) was dissolved in anhydrous dimethyl sulfoxide (DMSO, Carl Roth, Karlsruhe, Germany) and stored in a 10 mg/mL stock solution. 

### 2.12. General Procedures Relative to the Chemistry of Ami1_Mab_ Inhibitors

Proton (^1^H) NMR spectra were recorded on a Bruker Avance 300 (300 MHz for ^1^H), using Tetramethylsilane as an internal standard. Deuterated DMSO-d_6_ was purchased from Euriso-Top (Gif-Sur-Yvette, France). Chemical shifts are given in parts per million (ppm) (δ relative to the residual solvent peak for ^1^H). The following abbreviations are used: singlet (s), doublet (d), triplet (t), and multiplet (m). The instrument used was an Alliance Waters system [Alliance Waters 2695 (pump) and Waters 2998 (Photodiode Array detector)] and the column was a Waters XBridge C-18, 3.5 µm particle size (3.0 mm × 100 mm) (Waters, Saint-Quentin-en-Yvelines, France). Electrospray ionization mass spectrometry was recorded on a micromass ZQ 2000 (Waters, Saint-Quentin-en-Yvelines, France). Compounds 1 to 3 (purity >90% by liquid chromatography-mass spectrometry) came from the Institute Curie/CNRS chemical library (Orsay, France).

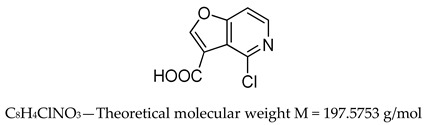
4-chlorofuro [3,2-c]pyridine-3-carboxylic acid (inhibitor 1) ^1^H NMR (300 MHz, DMSO-d_6_) δ 8.81 (s, 1H), 8.38 (d, *J* = 5.7 Hz, 1H), 7.85 (d, *J* = 5.7 Hz, 1H), LRMS (ESI-MS) *m/z* = 198.0 + 200.0 [M+H]^+^.
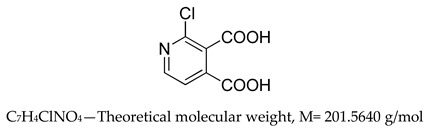
2-chloropyridine-3,4-dicarboxylic acid (inhibitor 2) ^1^H NMR (300 MHz, DMSO-d_6_) δ 14.07 (s, 2H), 8.62 (d, *J* = 5.0 Hz, 1H), 7.82 (d, *J* = 5.0 Hz, 1H). LRMS (ESI-MS) *m/z* = 202.0 + 204.0 [M+H]^+^.
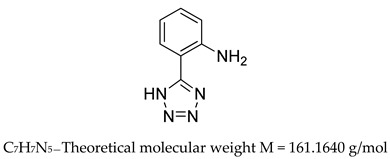
2-(1H-tetrazol-5-yl)aniline (inhibitor 3) ^1^H NMR (300 MHz, DMSO-d_6_) δ 9.09 (s, 3H), 7.84 (dd, *J* = 7.9, 1.2 Hz, 1H), 7.37 – 7.18 (m, 1H), 7.00 (d, *J* = 8.0 Hz, 1H), 6.80 (t, *J* = 7.2 Hz, 1H), LRMS (ESI-MS) *m/z* = 162.2 [M+H]^+^.


### 2.13. Transmission Electron Microscopy 

50 mL cultures were grown in 7H9 (supplemented with 10% OADC and 0.025% tyloxapol) and appropriate antibiotics to an OD600 = 0. Bacteria were harvested by centrifugation at 4000× *g* for 15 min and washed twice with 1 mL sterile PBS. Cells were fixed in 2.5% glutaraldehyde in PHEM (60 mM Pipes, 25 mM Hepes, 10 mM EGTA, 2 mM MgCl_2_, pH 6.9) buffer for 1 h at room temperature, post-fixed in OsO_4_ 1%/K4Fe(CN)_6_ 0.8% 1 h at room temperature then dehydrated in successive ethanol bathes (50/70/90/100%). The samples were then infiltrated with propylene oxide/epon 812 mixes, embedded in epon 812 and polymerized at 60 °C for 24 h. Then, 70 nm ultrathin sections were cut with a PowerTome XL ultramicrotome (RMC, Tucson, AZ, USA). Cuts were stained with 0.2% OTE (Oolong Tea Extract)/lead citrate and observed on a Tecnai G2 F20 (200 kV, FEG) TEM at the platform Plateau de Microscopie Electronique COMET, INM, Platform Montpellier RIO Imaging.

### 2.14. Zebrafish Infection Experiments 

Experiments were done with *M. abscessus* strains expressing tdTomato, obtained after transformation with the replicative plasmid pTEC27 [22]. Embryos were dechorionated using pronase (1 mg/mL final concentration) for 5 min at room temperature, followed by extensive washes in zebrafish water. At 30 h post-fertilization, embryos were anesthetized with 160 µg/mL tricaine (Sigma-Aldrich, Saint Quentin Fallavier, France) followed by caudal vein microinjection of 2–3 nL of bacterial suspension (approximately 100 CFU/nL). The size of the inoculum was verified a posteriori by injecting 2 nL of each bacterial suspension in sterile PBS and plated on 7H10 + OADC. To monitor embryo survival, infected larvae were transferred into 24-well plates (three embryos per well) and incubated at 28.5 °C. Embryos were monitored daily and dead embryos were determined based on the absence of a heartbeat and removed from the well.

### 2.15. Zebrafish Breeding and Ethics 

Zebrafish experiments were completed in accordance with the guidelines defined by the European Union for the use of laboratory animals. All animal experimentations were approved by the Direction Sanitaire et Vétérinaire de l’Hérault et Comité d’Ethique pour l’Expérimentation Animale de la région Languedoc Roussillon under the reference CEEA-LR-1145. Adult zebrafish were housed at the Centre National de la Recherche Scientifique, Montpellier, France. Zebrafish embryos were obtained by natural spawning and maintained at 28.5 °C in 60 µg/mL ocean salts. Zebrafish experiments were performed using the *golden* mutant line [32].

### 2.16. Statistical Analyses

Statistical analyses were performed on Prism 5.0 (Graphpad, La Jolla, CA, USA) and detailed in each figure legend. Macrophage experiments were analyzed using Student’s unpaired two-tailed *t*-tests. For zebrafish embryo survival curve analysis, log-rank (Mantel–Cox) statistical tests were used. 

### 2.17. Muramyl-Dipeptide Hydrolysis Assay

The reaction mixture was composed of 0.5 μL of Ami1 at different concentrations, 0.5 μL of MDP (InvivoGen, Toulouse, France) at 10 mg/mL and 4 μL of 1X PBS, and was incubated for 30 min at 37 °C. The whole reaction volume was spotted at the bottom of a silica gel TLC plate (10 × 10 or 20 × 20 cm) in 1 μL steps. Migration of the reaction products was done in a glass chamber using butanol–ethanol–water (ratio 50:32:18, *v/v/v*) as a mobile phase. After approximately 1 h, plates were removed, allowed to dry, sprayed with 2% ninhydrin solution (diluted in ethanol) and charred. MDP, the dipeptide, and *N*-acetylmuramic acid (Sigma-Aldrich, Saint-Quentin Fallavier, France) were used as internal standards. 

### 2.18. Compounds Screening by Thermal-Shift Assay 

The procedure was largely based on the protocol described by Niesen et al., [33]. Thirty-nine microliters of purified Ami1_Mab_ at a concentration of 80 µM were dispensed in 96-well PCR plates. One microliter of each compound at 10 mg/mL was added in each well. As a non-treated control and to determine Ami1_Mab_ melting temperature (T_m_), 1 µL of DMSO was added to one well with Ami1_Mab_. Thereafter, 40 µL of the fluorescent dye, SYPRO orange (Sigma-Aldrich, Saint-Quentin Fallavier, France), at a concentration of 10X (stock solution 5000X in 100% DMSO) was added. Plates were sealed and placed in a preheated qPCR machine (Agilent Mx3005) equipped with a Cy3 filter (excitation 545, emission 568). The heat cycle program was as followed: a 10 min step at 25 °C followed by an incremental of 1°/min step until the temperature reached 95 °C. After the run, the melting curves of the protein in each well were plotted using Graphpad Prism software. Curves were fitted using Boltzmann sigmoidal functions and the T_m_ was subsequently determined. A T_m_ value approximately of 2 °C to 5 °C higher than the DMSO control was considered as strong binding to the protein.

## 3. Results

### 3.1. Phenotypic Analysis of ami1 Deletion Mutant in the R and S Morphotypes

#### 3.1.1. *ami1_Mab_* Is Dispensable for *M. abscessus* Growth in Vitro

To investigate the role of Ami1_Mab_, a Δ*ami1_Mab_* deletion mutant was generated based on the development of a recent recombineering technique allowing the production of unmarked deletion mutants in *M. abscessus* [26]. Gene deletion was generated in both the S and R backgrounds. Proper gene inactivation was confirmed by PCR and sequencing analysis (Figure 1A).

The size of the amplicon ∆*ami1* mutants relative to the wild-type strain confirms the proper deletion of the gene in both the S and R morphotypes (Figure 1B). Complemented deletion strains were generated by cloning *ami1_Mab_* into the integrative pMV306 [34], yielding pMV306*::ami1_Mab_* and allowing gene expression under the control of the strong *hsp60* promoter. 

Growth curves of the Δ*ami1_Mab_* and the corresponding complemented strains were similar to those of their wild-type progenitors, suggesting that the inactivation of *ami1* does not affect the growth of *M. abscessus* in broth medium (Figure 1C). Overall, these results indicate that *ami1_Mab_* is dispensable for *M. abscessus* growth in vitro. 

#### 3.1.2. *ami1_Mab_* Deletion Mutants Do Not Present Division Defects

It was previously reported that 20% of the *M. smegmatis ami1* deletion mutants harbored an atypical cell division [11], although this observation was neither confirmed in an independent study [10] nor in a *M. tuberculosis ami1* deletion mutant [20]. We, therefore, investigated the shape and morphology of the *M. abscessus* S Δ*ami1* mutants by transmission electron microscopy (TEM) (Figure 1D). The R variant was not included in this analysis because of its high propensity to aggregate, making it difficult to obtain homogeneous bacterial suspensions, therefore rendering TEM processing very challenging. Analysis of approximately 500 individual WT, Δ*ami1_Mab_*, and Δ*ami1_Mab_* complemented cells failed to show morphological or cell division defects. Chain formation, typifying bacteria unable to divide properly, was not observed in Δ*ami1_Mab_*, thus differing from the chaining phenotype observed in *Helicobacter pylori* [35] or *E. coli* [17] strains in which single or multiple amidase-encoding genes were invalidated. Analysis of the size of a large number of single bacilli showed a slight but statistically-significant average shorter cell length for the WT (1.27 μm ± 0.018) and complemented strains (1.27 μm ± 0.017) as compared with the Δ*ami1_Mab_* mutant (1.34 μm ± 0.017) (Figure 1E). 

#### 3.1.3. Δ*ami1_Mab_* Is More Sensitive to β-Lactam Antibiotics and Detergent

We showed previously that inactivation of a dehydratase involved in cell wall biosynthesis was associated with increased susceptibility to detergents in *M. abscessus* [36]. In this study, we assessed whether *ami1* deletion affects cell wall integrity and thus susceptibility to sodium dodecyl sulfate (SDS) on solid medium. A moderate increase in sensitivity to SDS treatment at a concentration of 0.0025% was observed in Δ*ami1* as compared to the WT strain, particularly visible at the 10^−2^ dilution (Appendix A). The effect was more pronounced in the presence of 0.0037% SDS for the R variant. This low detergent sensitivity was partially restored upon complementation with *ami1_Mab_* (Appendix A).

As the Δ*ami1 M. smegmatis* and *M. tuberculosis* were more susceptible to antibiotics [11,20], we next inquired whether a similar antibiotic susceptibility profile is seen in Δ*ami1_Mab_*. The MIC of several β-lactam antibiotics (imipenem, cefuroxime, and cefamandole), known to target PG, were two times lower in the Δ*ami1_Mab_* strains compared to their respective parental progenitors (Table 1).

The wild-type sensitivity to these drugs was restored upon complementation. In contrast, the MIC of antibiotics that do not target PG (amikacin, clofazimine, and tigecycline) remained unaffected (Table 1). 

Taken together, these results suggest that although Δ*ami1_Mab_* mutants do not present major growth and cell division defects, they display a slightly-increased susceptibility profile to SDS and to β-lactam antibiotics, suggestive of a cell wall alteration. 

### 3.2. Analysis of the Virulence Phenotype of the M. abscessus ami1 Deletion Mutant

#### 3.2.1. Δ*ami1_Mab_* Is Not Impaired in Uptake and Persistence in Human Macrophages

The recent study by Healy et al., emphasized the importance of Ami1 for *M. tuberculosis* virulence, particularly for the successful establishment of chronic infection in mice [20]. Because immunocompetent mice rapidly clear the infection with *M. abscessus*, this model is not permissive to the establishment of a chronic infection [23]. We thus analyzed the virulence of the Δ*ami1_Mab_* mutants in the human THP-1 macrophages. Both S and R Δ*ami1_Mab_* strains were internalized by macrophages similarly to the parental strains, as judged from the CFU determination at day 0 (Figure 2A).

The capacity of the mutants to replicate inside macrophages was also slightly modified as CFU counts between day 1 and day 5 post-infection modestly increased as compared to the CFU of the corresponding WT strains (Figure 2A). Unexpectedly, the complemented strains overexpressing Ami1 showed an enhanced capacity to invade macrophages (up to 1-Log and 0.5-Log higher for the R and S variants at day 0, respectively) as compared to corresponding WT and mutant strains. However, the difference was less prominent at day 5 post-infection (Figure 2A). 

These results suggest that *ami1_Mab_* is not required for macrophage internalization and intracellular replication but that overexpression of Ami1_Mab_ confers an intracellular survival advantage to the bacilli. 

#### 3.2.2. Virulence of Δ*ami1_Mab_* Is Not Affected in the Zebrafish Model of Infection

Next, we assessed the virulence phenotype of Δ*ami1_Mab_* strains in zebrafish embryos, which are particularly suitable to study the chronology of *M. abscessus* infection [22]. WT, Δ*ami1_Mab_,* and complemented strains were first transformed with pTEC27 encoding tdTomato, allowing infection to be monitored in real-time by fluorescence microscopy. Bacterial suspensions (200–300 CFU) were microinjected in the caudal vein of zebrafish embryos at 30 h post-fertilization [22]. As reported previously, the S variant is predominantly avirulent in the embryos, with only 20% of embryo death at 12 days post-infection [22,37]. The survival curves of the embryos infected with Δ*ami1_Mab_* and the complemented strains were comparable to that of the WT (Figure 2B, left panel). In contrast, the R strains appeared more virulent with 70% larval death at 12 dpi, in agreement with previous reports [22] (Figure 2B, right panel). A slight but not statistically-significant killing delay was observed for the Δ*ami1_Mab_-* complemented strain (Figure 2B, right panel). These findings were corroborated with whole-embryo imaging, which failed to show infection foci in the embryos infected with the different S-derivative strains (Figure 2C, left panel), while robust infection occurred in the embryos challenged with the R derivative strains, typified by the presence of massive infection and important bacterial burdens within the central nervous system (Figure 2C, right panel). Together, these results suggest that Ami1*_Mab_* is not required for *M. abscessus* virulence in zebrafish.

### 3.3. Biochemical and Structural Characterization of Ami1_Mab_

#### 3.3.1. Ami1_Mab_ Is an Active PG Hydrolase

The biological function of Ami1 in *M. smegmatis* and *M. tuberculosis* underscores its requirement for growth, cell division, and virulence. Ami1 appears, therefore, as a potential drug target to exploit in mycobacteria. The macrophage infection results highlight the intracellular advantage conferred by overexpression of Ami1_Mab_ in *M. abscessus*. Comforted by these studies, we initiated biochemical studies, searched for specific inhibitors, and characterized the structure of Ami1_Mab_. 

Ami1_Mab_ was purified without its first 40 amino acids in *N*-terminus, predicted to correspond to the signal peptide required for secretion. It was purified as a recombinant protein expressed in *E. coli* following a three-step purification procedure, yielding a highly pure and homogeneous protein preparation (Figure 3A).

The hydrolytic activity of pure Ami1_Mab_ was next monitored by thin-layer chromatography (TLC) and using the commercial *N*-acetylmuramyl-l-alanyl-d-isoglutamine substrate, also known as muramyl-dipeptide (MDP). If the amidase is active on MDP, *N*-acetylmuramic acid and l-Ala-d-IsoGln are expected to be released (Figure 3B). Substrate and reaction products could be visualized following separation by TLC, spraying with ninhydrin, and charring. We first monitored the hydrolysis of MDP using a range of protein concentrations (from 2 to 32 µM). MDP was rapidly hydrolyzed by Ami1_Mab_, indicating that it is an active PG hydrolase that cleaves the amide bond between the *N*-acetylmuramic acid and l-Ala (Figure 3C). Since Zn^2+^ is an essential divalent ion for PG amidases, we next assessed the dependence of Ami1_Mab_ for this metal ion. Incubating Ami1_Mab_ with the chelating agent EDTA prior to the reaction almost completely abolished the enzymatic activity (Figure 3D). These results demonstrate that, like Ami1_Mtb_ (Figure 3D), [15,16] Zn^2+^ is required for catalysis of Ami1_Mab_.

#### 3.3.2. Thermal-Shift-Based Drug Screening Identified Three Potent Inhibitors of Ami1_Mab_

Despite being a robust assay to monitor PG hydrolytic activity, the TLC-based assay is not compatible with a large screening program to identify specific inhibitors of amidases. To overcome this hurdle, we employed thermal-shift assay (TSA) [38,39] to screen a large chemical library. This technique is based on the principle that if a small molecule binds to the protein, the melting temperature (Tm) of the protein is modified. The Tm can easily be followed by adding a fluorescent dye, SYPRO orange, that binds to hydrophobic patches of a protein that are exposed after melting. Monitoring of the signal can be performed by standard real time-PCR. It is noteworthy that this method does not necessarily select for direct enzyme inhibitors but identifies molecules that can stabilize or destabilize the protein of interest. 

A TSA approach was developed in a 96-well plate format to screen the 9200 compounds of the Curie-CNRS library. The entire screen consisted of 115 plates, each containing 80 chemicals as well as a non-treated control and Ami1_Mab_ plus 5% dimethyl sulfoxide (DMSO). Compound selection was based on Tm values ranging from 2 to 5 °C higher than the Tm of Ami1_Mab_ plus DMSO control. Application of this threshold allowed the identification of 654 compounds. All selected compounds were then assessed individually (at a concentration of 0.125 mg/mL) for their ability to inhibit Ami1_Mab_ activity using the TLC-based assay. This narrowed down our initial screen to 54 compounds. A third round of selection was next applied, keeping only compounds inhibiting the enzyme activity at a concentration below 1 mM. This led to the selection of only three potential hits.

#### 3.3.3. Biochemical Characterization of Ami1_Mab_ Inhibitors

Among these three hits, two belong to the pyridine chemical group while the third one is a tetrazole-derived compound. Compound **1** is a 4-chlorofuro [3,2-c]pyridine-3-carboxylic acid while compound **2** is a 2-(1H-1,2,3,4-tetrazol-5-yl)aniline and compound **3** is a 2-chloropyridine-3,4-dicarboxylic acid (Figure 4A).

The validity of these hits was subsequently reassessed by TSA with respect to their capacity to increase the Tm of Ami1_Mab_. The three compounds increased the Tm of Ami1_Mab_ by 5.5, 2.4, and 3.5 °C, respectively, when compared with Ami1_Mab_ incubated with DMSO (Tm= 45 °C) (Figure 4B). 

The capacity of the three compounds to inhibit the MDP hydrolytic activity of Ami1_Mab_ was determined in a concentration-dependent manner using the TLC-based activity assay. Compound **1** displayed the highest inhibitory effect, starting at 150–200 μM, followed by compound **3** between 250–500 μM, and compound **2**, which showed a moderate inhibitory activity starting at 800 μM (Figure 4C). 

Since two of the inhibitors possess carboxylic groups, we addressed whether the inhibitory effect was not due to a pH change during the reaction. The Ami1_Mab_ activity was thus tested under various acidic conditions, resulting in an active protein until pH 2 was reached. In parallel, products and substrates of the reaction were separated by TLC under the same pH range to ensure that the observed inhibition was not due to a TLC migration artifact dependent on the pH (Figure 4D). 

These biochemical data indicate that all three compounds inhibit the Ami1_Mab_ activity in vitro. To enlighten the mode of inhibition of these compounds, structural studies of Ami1_Mab_ were undertaken.

#### 3.3.4. Biochemical Characterization of Ami1_Mab_ Inhibitors

The crystal structure of Ami1_Mab_ was solved at a high resolution in its free form and also bound to l-Ala-d-IsoGln, one of the end-product of the reaction (Table 2).

In addition, to have a complete structural view and to compare the Ami1 structures from *M. tuberculosis, M. smegmatis,* and *M. abscessus,* we have also solved the crystal structure of Ami1_Msm_ at a high resolution (Table 2). 

Ami1_Mab_ and Ami1_Msm_ displayed an overall globular structure comprising a central β-sheet (β1, 4, 5, 6, 7, 8) surrounded by strands β2 and β3 forming an antiparallel β-sheet, six α-helices, and one 3_10_ helix (η1) (Figure 5A).

The catalytic zinc is coordinated by the highly-conserved triad which constitutes the amidase_3 motif signature, namely H25, E60, and H115 in Ami1_Mab_ and H51, E86, and H141 in Ami1_Msm_ (Figure 5A, left and middle panels). Ami1_Mab_ shares 68% and 65% primary sequence identity with Ami1_Msm_ and Ami1_Mtb,_ respectively, while Ami1_Msm_ and Ami1_Mtb_ possess 79% identity_._ All three structures of the *apo* forms are very similar since the superposition of Ami1_Mab_ with Ami1_Msm_ and Ami1_Mtb_ (PDB id: 4LQ6) leads to r.m.s.d. values of 0.46Å and 0.55Å, respectively, while Ami1_Msm_ and Ami_Mtb_ display a r.m.s.d. value of 0.51Å. 

The co-crystal structure of Ami1_Mab_ bound to l-Ala-d-IsoGln was also solved at a high resolution. The Fo-Fc simulated-annealed OMIT map attests to the presence of the dipeptide in the active site of the protein (Figure 5B). This structure represents the enzyme state after the hydrolysis of MDP. The dipeptide binds close to the Zn^2+^ ion (Figure 5C). The l-Ala residue of the dipeptide is tightly bound through water-mediated H-bonding established with the side chains of Asp117 and Arg173, itself stabilized by the side chain of Asp175. Glu190 interacts with the dipeptide ligand and this residue was shown to be critical for the in vitro and in vivo activity of Ami1_Mtb_ [15,16,20]. The carboxylic group of d-IsoGln is contacted by a salt-bridge interaction with Lys45 as well as by the main chain of Ala177 and Gly178. The side chain of Gln48 finalizes the interaction with the amide group of d-IsoGln. All these interactions are conserved in the dipeptide-bound Ami1_Mtb_ structure (PDB id: 4M6G) (Figure 5D) and presumably also in Ami1_Msm_ since all the residues interacting with the ligand in Ami1_Mab_ and Ami1_Mtb_ are conserved in Ami1_Msm_ (Figure 5E). We noticed, however, a slight difference in ligand binding in the structures of Ami1_Mab_ and Ami1_Mtb_ bound to l-Ala-d-IsoGln. In the latter, the NH_2_ group of the Ala residue of the dipeptide is close to Zn^2+^ and only interacting with Glu200 (Figure 5C,D). In Ami1_Mab_, the amine group is interacting with the side chains of Asp117, Glu190 and, Arg173 (Figure 5C). 

Overall, these findings demonstrate that Ami1_Mab_ is a functional PG hydrolase that shares a very strong structural conservation with Ami1_Mab_ and Ami1_Mtb_.

#### 3.3.5. In Silico Docking Allows Predicting That Ami1_Mab_ Inhibitors May Compete with Substrate Binding

We did not succeed in solving the co-crystal structure of Ami1_Mab_ bound to any of the three selected inhibitors. The relatively-low potency of the inhibitors is in favor of a low affinity of these inhibitors for the amidase and might explain our vain attempt at obtaining the co-structures. As an alternative, in silico docking was performed to gain insight into their potential mechanism of inhibition. To facilitate the search for the potential inhibitor’s binding-site, the Ami1_Mab_ crystal structure was probed with the CASTp server to identify cavities in the protein structure [40]. This analysis led to the identification of a single cavity of a volume of 72 Å^2^ comprising the active site of Ami1_Mab_. In silico docking by defining a search region in this area was performed with the AutoDock Vina software [31]. The three inhibitors could be docked into the Ami1_Mab_ active site with similar binding energy (ΔG°) (Figure 6).

Inhibitor 1 (ΔG° = −6.5 kcal/mol) appears to be able to occupy the active site by interacting with the main chain of Ala177 and Gly178 through its carboxylic group and a weaker H-bonding with the Lys45 side chain. The nitrogen of the pyridine moiety is interacting with the side chain of Asp22, His25, Gln48, and His115 interact as well by Van der Waals interactions. Inhibitor 2 (ΔG° = −6.0 kcal/mol) interacts by H-bonding with Lys45 and Gln48 as well as with His25 and His115 by weaker contacts. Inhibitor 3 could be docked with a ΔG° = −6.4 kcal/mol, where multiple interactions involve the two carboxylic groups. The carboxylate in position 3 interacts with the side chains of Gln48, Arg99, and the main chain of His25, while the carboxylate in position 4 is able to contact the side chains of Asp22, Ser113, and Lys186. Ile96 side chain is in close vicinity with the chloride in position 2.

The proposed docking poses strongly suggest that the three inhibitors are able to occupy the active site in a manner similar to the dipeptide seen in the crystal structure. The carboxylate group of inhibitor 1 is interacting with the same residues Lys45, Ala177, and Gly178 involved in binding of the d-IsoGln carboxylate. The docking study supports the view that these inhibitors might act by competing with the binding of the PG peptide stem. 

## 4. Discussion

The PG biosynthesis pathway has long been overlooked as an interesting drug target for subsequent drug developments against mycobacterial diseases, largely due to the fact that most β-lactam antibiotics inhibiting transpeptidation show poor activity against mycobacteria. This is explained by the presence of specific secreted β-lactamases, which neutralize these antibiotics. However, targeting PG biosynthesis in mycobacteria has recently regained interest after the discovery of β-lactamases inhibitors which, together with β-lactam antibiotics, can successfully inhibit growth of *M. tuberculosis* [41] and *M. abscessus* [42,43].

PG remodeling in mycobacteria and particularly PG hydrolytic enzymes has been more thoroughly investigated in recent years [44]. The NlpC/P60 endopeptidases RipA, RipB, RipC, and RipD have been widely scrutinized [7,20,45,46,47,48,49,50]. The functional redundancy and the non-essential individual roles of RipA and RipB hydrolyzing the PG peptide stem between the 2nd and 3rd amino acid were established both in *M. smegmatis* [50] and in *M. tuberculosis* [20]. Redundancy of PG amidases has also been proposed since single or multiple deletion of amide-encoding genes (*ami2*, *ami3,* and *ami4*) in *M. smegmatis* is not detrimental for growth [10]. The major septal PG hydrolase RipA can sustain the amidase Ami1 PG hydrolytic activity in *M. tuberculosis* [20]. However, despite the fact that mycobacterial amidases share redundant functions and that *ami1* is not essential for growth or cell division of *M. tuberculosis,* Ami1 is required for the persistence of the bacilli within the host [20]. This contrasts with studies reporting that Ami1 is essential for cell separation and that inactivation of the corresponding gene is associated with atypical cell division phenotypes [11,13]. 

Apart from *M. smegmatis*, little is known about PG-hydrolyzing enzymes in other non-tuberculous mycobacteria. To fill this gap, we investigated the role of *ami1* in the human pathogen, *M. abscessus.* We showed that *ami1_Mab_,* as reported in *M. smegmatis* and *M. tuberculosis,* is dispensable for cell growth in the R and S variants. Consistent with observations in *M. tuberculosis*, *ami1_Mab_* is also not essential for cell division in *M. abscessus*. We also showed that *ami1_Mab_* is neither required for invasion nor for persistence in macrophages while overexpression of *ami1_Mab_* partially increased the uptake and intracellular survival of the bacilli in THP-1 macrophages. In *M. smegmatis,* deletion of *ami1* increases type 3-3 crosslinks as compared to the WT strain [11]. In *M. abscessus,* changes in PG cross-link ratio and type of crosslinking is correlated with the growing phase. PG of exponentially-growing *M. abscessus* has more cross-links characterized by an increasing ratio of 3-3 cross-links [51]. As the deletion of *ami1* in *M. smegmatis* increases the 3-3 crosslinks ratio [11] one could propose that overexpression of *ami1_Mab_* might trigger a decrease in 3-3 and consequently an increase in 4-3-type cross-links, which may confer, for yet unidentified reasons, intracellular advantage. This, however, remains to be further investigated in future studies. The evidence of *ami1_Mab_* being involved in PG remodeling was also supported by a slight increase in the susceptibility to *β*-lactam antibiotics in Δa*mi1_Mab_*. MICs for non-targeting PG antibiotics were not modified, suggesting that *ami1* deletion triggers specific cell wall modification in PG rather than modifications of the permeability of the cell wall. The cell wall impairment in Δ*ami1_Mab_* is also supported by a modest increased sensitivity to detergent. 

Infections studies in zebrafish embryos and macrophages suggest that Δ*ami1_Mab_* is as virulent as its WT progenitor. In light of the recent studies related to Ami1 from *M. tuberculosis* and *M. smegmatis*, this work on Ami1_Mab_ emphasizes functional differences regarding the contribution of Ami1 from the three species regarding growth, invasion, and persistence. These findings are particularly unexpected considering that all three mycobacteria possess the four amidase-encoding genes and that the corresponding orthologues share a highly-conserved primary sequence, suggesting that these functional differences are unlikely connected to their enzymatic activity per se. Super-resolution microscopy analyses unraveled differences in the spatiotemporal synthesis of PG in *M. smegmatis* and *M. tuberculosis* [52], which may explain the functional divergence of Ami1 between the different mycobacterial species. 

Growth conditions and bacterial environment also appear crucial in modulating PG remodeling enzymes as reported for *ripA* deletion mutant which is only impaired in cell division under acid stress conditions (8, 37). A possible explanation for the difference of Ami1 contribution may arise from the respective lifestyles of the environmental *M. abscessus* as opposed to the human-specific *M. tuberculosis* pathogen. In this context, it is noteworthy that the obligate intracellular *Mycobacterium leprae,* carrying a reduced genome, encodes for only two amidases, namely Ami1 (ML2131) and Ami2 (CwlM, ML2704). This supports the view that Ami1 and Ami2 are required for the intracellular mycobacterial lifestyle and were conserved in *M. leprae* due to the selection pressure. Additionally, transcriptomic analyses showed up- and down-regulation of multiple genes in *M. abscessus subspecies massiliense* when comparing planktonic, intra-macrophage, and intra-amoebal growth [53]. These included genes from the PG metabolic pathway found to be down-regulated in macrophages as compared with planktonic growth conditions [53]. 

Biochemical and structural characterizations of Ami1_Mab_ unambiguously demonstrate that it is an active PG amidase, very closely related to Ami1_Msm_ and Ami1_Mtb_. A noticeable difference in dipeptide binding was observed between Ami1_Mab_ and Ami1_Mtb_. This could be explained by the fact that the Ami1_Mab_ structure was co-crystallized with the dipeptide while the structure of Ami1_Mtb_ which was generated by crystallization with MDP, likely hydrolyzed during the crystallization process and only the dipeptide part was visible in the structure. In the Ami1_Mab_ co-structure, the peptide is further away from the catalytic Zn^2+^. This state might, therefore, represent a post-hydrolysis state i.e., a state where the product is about to be released from the enzyme.

This study describes the first inhibitors of this class of enzymes, which may represent a first step towards the design of more potent inhibitors. Further work through structure–activity relationship approaches is expected to improve the properties of these inhibitors. 

In summary, this work represents the first study describing a PG hydrolytic enzyme in *M. abscessus*. Further work is needed to (i) explore the role of Ami1 in *M. abscessus* and its intricate relationships with other amidases/endopeptidases such as RipA and (ii) study its regulation under various stress conditions.

## Figures and Tables

**Figure 1 cells-09-02410-f001:**
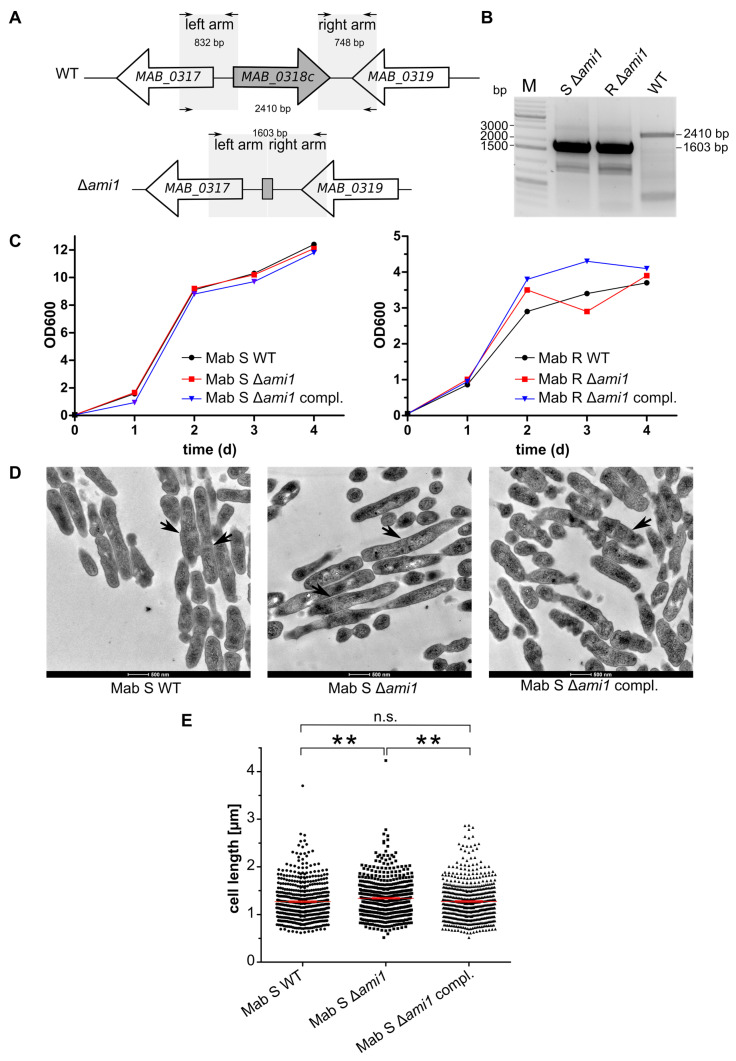
Generation and phenotypic analyses of Δ*ami1_Mab_*_._ (**A**) Schematic representation of the genomic region around *ami1_Mab_ (MAB_0318c)* in the parental (WT) and Δ*ami1*_Mab_ strains of *Mycobacterium abscessus*. The size of the PCR amplicons used for genotyping the Δ*ami1_Mab_* mutants is indicated. (**B**) PCR profile confirming the proper deletion of *ami1_Mab_* in the mutant strains. The PCR products of 1.6 kb were amplified from Δ*ami1_Mab_* genomic DNA from rough (R) and smooth (S) variants while a band at 2.4 kb is expected for the parental (WT) strain. (**C**) In vitro growth curves at 37 °C of the *M. abscessus (Mab)* WT, Δ*ami1_Mab_*, and complemented strains, in the R and S variants, respectively. Bacteria were grown in 7H9 broth supplemented with 10% oleic acid-albumin-dextrose-catalase (OADC) and 0.025% tyloxapol. The experiment is one representative of three replicates. (**D**) Transmission electron microscopy (TEM) of the *M. abscessus* WT, Δ*ami1_Mab_*, and complemented strain in the S morphotype. Septa are indicated by black arrows. (**E**) Cell length measurements from the TEM data. Mean cell length = 1.27 μm for *Mab* S WT (*n* = 518); mean cell length = 1.34 μm for Δ*ami1_Mab_* (*n* = 557); mean cell length = 1.27 μm from the complemented strain (*n* = 576), the scale bar represents 500 nm. For statistical analysis, an unpaired *t*-test with Welch’s correction was performed. *p* = 0.0035 for WT vs. Δ*ami1_Mab_* and *p* = 0.0053 for Δ*ami1_Mab_* vs. Δ*ami1_Mab_* complemented (compl.) n.s. stands non statistically significant and ** stands for *p* < 0.01.

**Figure 2 cells-09-02410-f002:**
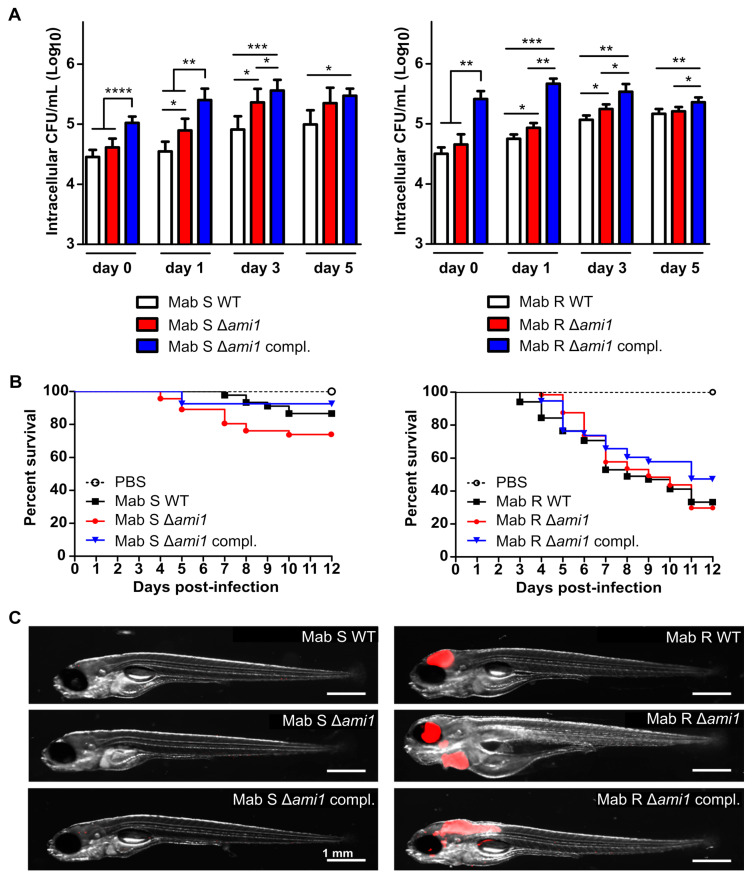
Assessment of Δ*ami1*_Mab_ in human macrophage and zebrafish larvae. (**A**) *M. abscessus*-infected THP-1 macrophages with amikacin maintained at 50 μg/mL in the RPMI medium were lysed from day 0 to day 5 for CFU enumeration. Bacteria were plated at day 0 (2–4 h phagocytosis followed by a 2 h amikacin treatment at 250 μg/mL) day 1, 3 and day 5. Histograms and error bars represent means and standard deviations calculated from three independent experiments. For statistical analysis, the unpaired *t*-test with Welch’s correction was applied. *, **, ***, and **** stand for *p* < 0.1, *p* < 0.01, *p* < 0.001, and *p* < 0.0001, respectively. Left and right panels are for *M. abscessus* S and R variants, respectively. The *M. abscessus* S and R wild-type strains used as controls were not transformed with pMV306. (**B**) Survival curves of zebrafish embryos infected with the different *M. abscessus* strains. The data shown are corresponding to a pool of data from three independent experiments. For each experiment, 20 embryos were injected with 200–300 CFUs, respectively. Left and right panels are for *M. abscessus* S and R variants, respectively. The *M. abscessus* S and R wild-type strains used as controls were transformed with pTEC27 but not transformed with pMV306. (**C**) Representative images of infected zebrafish embryos at 5 days post-infection. Scale bar represents 1 mm. Infection foci are displayed in red.

**Figure 3 cells-09-02410-f003:**
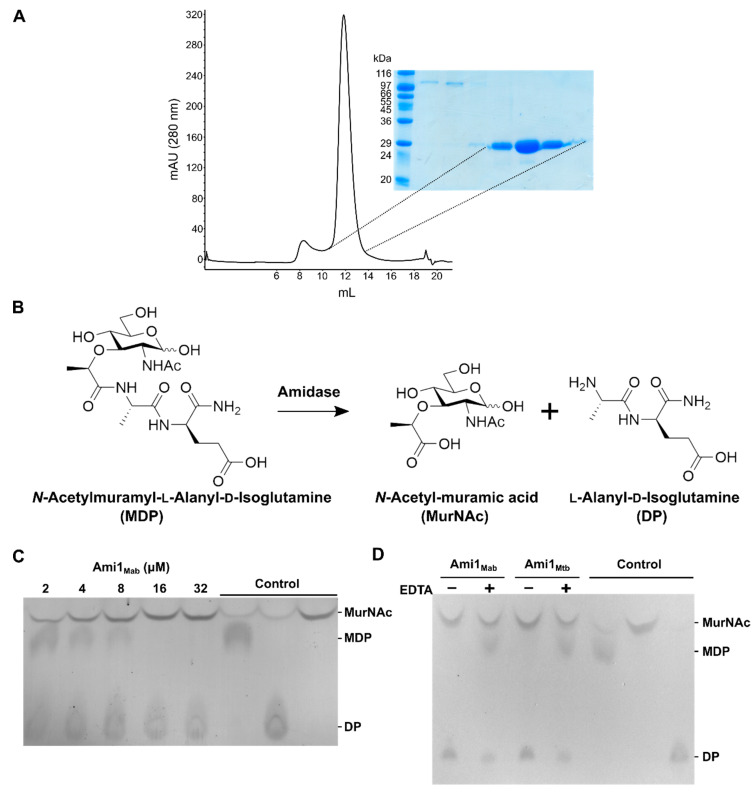
Biochemical characterization of Ami1_Mab_. (**A**) Ami1_Mab_ elution profile on Superdex 75 10/300 GL size-exclusion chromatography column. The denaturating polyacrylamide electrophoresis gel attests to the high protein purity after three chromatography steps. (**B**) Schematic of the muramyl-dipeptide hydrolysis assay. Ami1_Mab_ cleaves muramyl-dipeptide (MDP) to yield *N*-acetylmuramic acid (MurNAc) and l-Ala-d-IsoGln (dipeptide). (**C**) The thin-layer chromatography (TLC)-based activity assay revealed a correlation between the disappearance of MDP and the concentration of Ami1_Mab_. Pure MDP, MurNAc, and dipeptide (DP) were loaded on the right part of the TLC plate as migration controls. TLC was revealed by ninhydrin and charring. (**D**) The enzymatic activity of Ami1 is Zn^2+^-dependent. Activity of both Ami1_Mab_ and Ami1_Mtb_ is inhibited after incubation with EDTA.

**Figure 4 cells-09-02410-f004:**
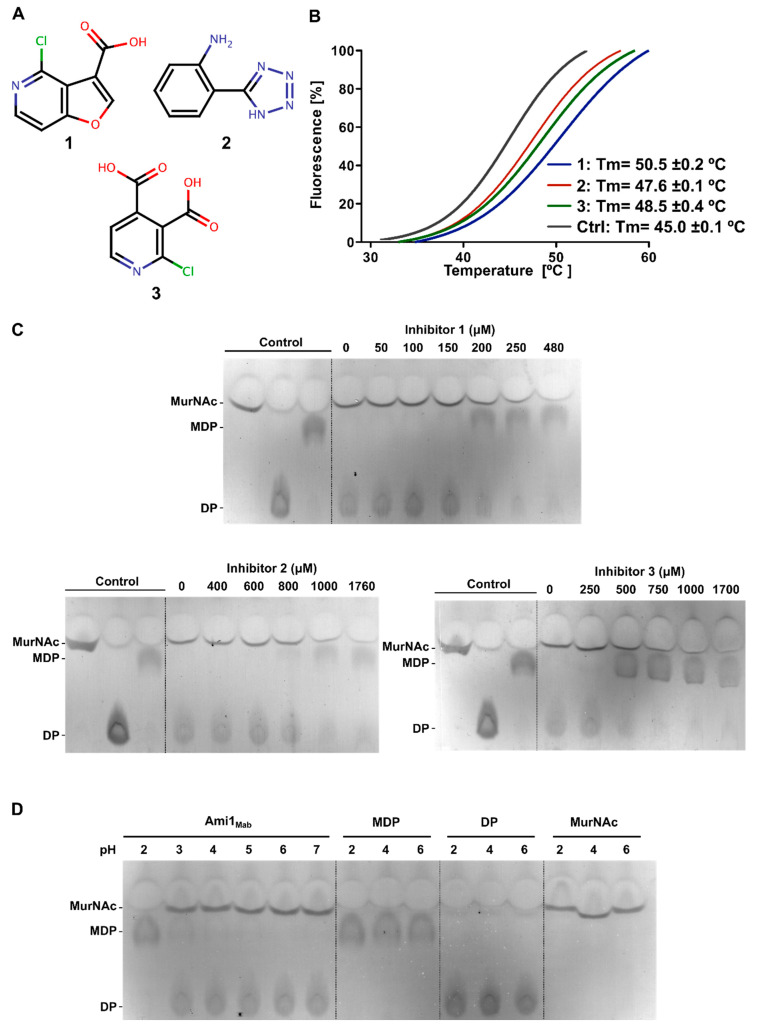
Characterization of Ami1_Mab_ inhibitors. (**A**) Chemical structure of the three compounds selected by thermal-shift assay. Chloride atom is in green, carbon in black, oxygen in red and nitrogen in blue (**B**) Melting curves of Ami1_Mab_ in the presence of compound **1** (blue) at 244 μM, compound **2** (red) at 879 μM, and compound **3** (blue) at 845 μM as compared with protein incubated with 5% DMSO (black). (**C**) Assessment of Ami1_Mab_ inhibition by compounds 1, 2, and 3 by TLC. (**D**) Evaluation of the Ami1_Mab_ activity over an acidic pH range.

**Figure 5 cells-09-02410-f005:**
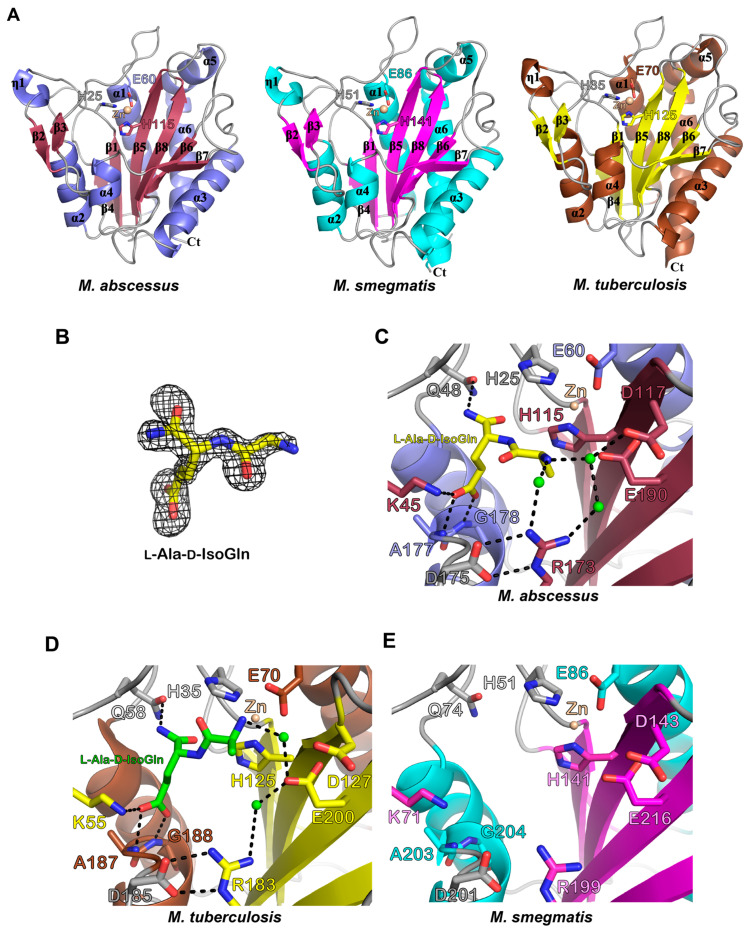
Structural comparisons of the Ami1_Mab_, Ami1_Mtb_, and Ami1_Msm_. (**A**) Overall crystal structure of Ami1_Mab_ (left)_,_ Ami1_Msm_ (middle) and Ami1_Mtb_ (PDB id: 4LQ6) (right). Each structure is shown as a cartoon representation and is colored differently and according to its secondary structures. α, β, and η indicate alpha-helices, beta-strands, and 3_10_ helices, respectively. The catalytic zinc ion (Zn^2+^) appears as a wheat-colored sphere. (**B**) Fo-Fc simulated-annealed OMIT map. The electron density map as seen in the Ami1_Mab_:dipeptide co-crystal structure and surrounding l-Ala-d-IsoGln (yellow sticks) is displayed as a grey mesh and contoured at 3 σ level. (**C**) Close up of the active site of Ami1_Mab_ as seen in the Ami1_Mab_: dipeptide co-crystal structure. l-Ala-d-IsoGln (yellow sticks) tightly interacts by H-bonding and salt-bridge as illustrated by the dashed lines. Water molecules appear as green spheres. (**D**) The figure shows the molecular interactions seen in the co-crystal structure of Ami1_Mtb_ bound to dipeptide (PDB id: 4M6G). (**E**) Active site of the Ami1_Msm_.

**Figure 6 cells-09-02410-f006:**
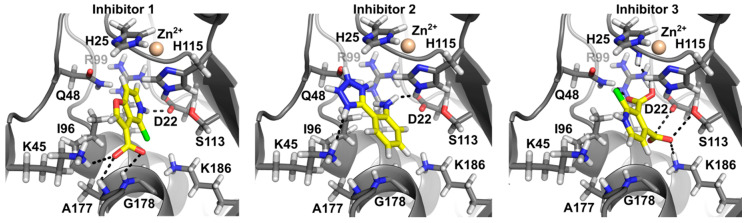
In silico docking of the three inhibitors. The figure depicts the best docking poses of the three compounds. All three inhibitors appear able to bind in the Ami1_Mab_ active site with similar binding energies. Inhibitors are displayed in stick representation. Nitrogen atoms are in blue, oxygen in red, hydrogen in white and carbon atoms in yellow for the ligand and grey for the protein.

**Table 1 cells-09-02410-t001:** MIC determination

	*M. abscessus* S WT	*M. abscessus* S Δ*ami1*	*M. abscessus* S Δ*ami1* Compl.
Imipenem	4	2	4
Cefoxitine	16	16	16
Cefuroxime	187–375	93	187
Cefamandole	2000	500–1000	2000
Amikacin	16	16	16
Clofazimine	0.78	0.78	0.78
Tigecycline	4	4	4

MIC in μg/mL of the different antibiotics were determined as described in the Materials and Methods, Section 2.8.

**Table 2 cells-09-02410-t002:** Crystallographic data collection and refinement statistics.

	Ami1_Mab_	Ami1_Mab_ l-Ala-d-IsoGln	Ami1_Msm_
PDB accession	7AGL	7AGO	7AGM
Beamlime	ESRF-ID30B	SLS-PXIII-X06DA	SLS-PXIII-S06DA
Wavelength	0.979	1	0.979
Resolution range	38.4–1.6 (1.65–1.6)	42.8–1.7 (1.76–1.7)	46.7–1.35 (1.39–1.35)
Space group	P 41 21 2	P 41 21 2	P 1 21 1
Unit cell	85.92 85.92 73.66 90 90 90	85.74 85.74 75.63 90 90 90	41.85 68.84 63.72 90 91.866 90
Total reflections	228,475 (22,850)	824,584 (80,940)	526,861 (52,052)
Multiplicity	6.4 (6.3)	26.1 (26.1)	6.7 (6.6)
Completeness (%)	96.7 (99.2)	99.8 (100)	99.9 (99.9)
Mean I/sigma(I)	11.1 (1.3)	18.8 (2.4)	16.0 (1.3)
Wilson *B*-factor	22.4	20.9	15.4
R-meas	0.094 (1.25)	0.131 (1.29)	0.069 (1.47)
CC1/2	0.99 (0.48)	0.99 (0.85)	0.99 (0.61)
Reflections used in refinement	35,744 (3601)	31,542 (3107)	79,191 (7884)
Reflections used for *R*-free	2000 (201)	2000 (197)	3960 (394)
*R*-work	0.168 (0.294)	0.162 (0.256)	0.155 (0.285)
*R*-free	0.195 (0.329)	0.186 (0.307)	0.180 (0.319)
Number of non-hydrogen atoms	1831	1857	3672
Macromolecules	1598	1628	3218
Ligands	1	15	2
Solvent	232	242	452
Protein residues	212	214	434
RMS (bonds, Å)	0.007	0.006	0.005
RMS (angles, °)	0.89	0.79	0.77
Ramachandran favored (%)	97.14	97.17	97.91
Ramachandran allowed (%)	2.86	2.83	2.09
Ramachandran outliers (%)	0.00	0.00	0.00
Rotamer outliers (%)	1.18	1.18	0.30
Clashscore	0.63	1.55	1.72
Average *B*-factor (Å^2^)	27.6	22.6	23.2
Macromolecules	25.8	20.5	21.6
Ligands	20.4	33.4	18.3
Solvent	40.1	35.8	34.3

Values in parentheses are for the last resolution shell.

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
