# Peer review of "Functional Characterization of the N-Acetylmuramyl-l-Alanine Amidase, Ami1, from Mycobacterium abscessus"

_cells, 2020, doi:10.3390/cells9112410_

Round 1
Reviewer 1 Report
The study by Küssau et al is focused on the function of the Ami1, a PG amidase from M. abscessus.
The topic is relevant since PG enzymes have been poorly studied in M. abscessus. The presentation is appropriate and accurate; the rational design is comprehensible. The obtained results are of interest and, even if not conclusive, they describe Ami1 in M. abscessus and three Ami1 inhibitors.
The quality of writing is fine, there are minor spell checks required or editing issues.
L 138-139, 144-145, 150-151 Please, the oligo style should be made uniform.
L223 Add a reference like Palomino, et al, 2002
Fig. 1 C Is the experiment one representative of three replicates?
L 376 ami1 should be in Italic.
L389-390 the mean values are different from those in Fig 1 E
L 295 check 1h RT abbreviation
L395-396 at both concentrations
L403 Remove “was”
L411-416 Remove this part, it is already present in Materials and methods section
Paragraphs 3.3.1 and 3.3.2 Please, “FIGURE” should be made uniform.
L514 Could you insert the concentration, please?
Reviewer 2 Report
Understanding the function of peptidogylcan - modyfying enzymes like PG-amidase1 in mycobacterial infections have an major impact on identifying and defining new targets for antimycobacterial treatment. The present study shows a comprehensive and extensive analysis of amidase 1 of M. abscessus.
Minor comments/suggestions
- l1: "molecular biology" in capital letters
- Material and Methods:
- Section 2.1 – 2.2.: all sequences should be provided in an extra table in the supplement
- Section 2.3: Why are you using Tyloxapol instead of Tween80? Tyloxapol has a significant impact on „clumping“ behaviour and modulates growth profiles, differences between the rough and smooth morphotype are maybe artefacts in this assay
- Results:
- Figure 1C: please show SEM, OD of the rough morphotype is significant lower than the smooth morphotype. Are CFU data are avaiable?
- ll 411 – 416: It is redundant description of section 2.8 with small changes i.e. inoculum (5x106 CFU/ml vs. 4x106 CFU/ml) or drug dilution (2 µl in 198µl vs 2µl in 200µl)
- Figure 2A: How long was phagocytosis? (Material and Methods: 4h, here 2-3 h)
- ll 506 – 514 should be moved to „Material and Methods“
- Table 2: Harmonise layout to table 1
- Figure 5A: figure legend is not fitting to the figure sequence (Ami1Mtb is the right panel and not the middle one)
- Check paper for Ami1, sometimes „1“ is missing e.g l. 568, 569, 578, 584,589, 683 ….
Reviewer 3 Report
The manuscript is generally very interesting. It uses a large number of experimental techniques to characterize the Ami1 protein from Mycobacterium abscessus and to evaluate this protein as a potential drug target.
Specific comments:
1.In the section describing protein purification for crystallography purposes, the authors mention the purification of Ami1 from M.tuberculosis, but this protein is not being implemented for any assays/ inhibitor screening.
2.Were the mutant strains lacking Ami1 confirmed by any alternative technique rather than just PCR? Southern blotting or global sequencing would be much more accurate.
3.Are the results of the TLC experiment of sufficient quality to be quantifiable? It would probably be helpful for the Reader to see the percentage of the MDP hydrolysis at a given concentration and upon treatment with EDTA as well.
4.While the search for potential inhibitors was dealt with cleverly by using the thermal shift assessment to prescreen the drug library, it only led to finding rather very weak inhibitors of Ami1Mab. Since the authors had an Msm and Mtb proteins on hand as well, wouldn't it be sensible to also assess their activity against the other two counterparts in similar assays?
5.Regarding the structural data, it is surprising to see a difference in binding to the D-IsoGln ligand between the Mab and Mtb since the residues constituting the active site are highly conserved.
6.If there is a slight change in the binding to the Mab vs Mtb proteins the authors should have checked if the three inhibitors are active and possibly more potent towards the Mtb protein instead.
7.Do the authors have any data on the Ami1Mab being involved in peptidoglycan recycling, rather than being involved in cell division and cell cycle progression? Increased recycling could potentially be beneficial for intracellular bacteria, thus promoting survival and growth in your infection model.
8.The SDS sensitivity assay could have been repeated on disks, measuring the zone of inhibition as well, to make it more clear and give some sense of digitalization. Lysozyme sensitivity (even with human lysozyme commercially available) could also have been tested for the evaluated strains to complement that and help explain the potential difference in survival during infection.
Minor editing issues were spotted:
line 92 are increasingly globally
line 95 but however
Reviewer 4 Report
PG amidase Ami1 was characterized in this paper through virulence test using THP-1 and ZF model of infection, an enzymatic assay using TLC, and searching Ami1Mab inhibitor using hits that were screened library.
The manuscript was well written however requires some answers to the questions below.
- Line 122 : Author constructed pUX1-KatG plasmid for M. abscessus. I wonder whether it was easy to select colonies that resistant to Isoniazid? Because M.abscesus is intrinsically resistant to isoniazid. Please carefully explain the background for this?
- Move line 412-416 into the Materials and method section.
- Please described ethics committee permission number for ZF experiment.
- For Figure 2, I really do not understand why complemented strains overexprssing Ami1 exhibited higher intracellular growth in comparison with WT. Did you use control strain with M. abscessus CIP104536T for THP-1 macrophages experiment or M. abscessus CIP104536T containing pMV306 as a control? Please clarify it for and Figure 2a, b and c respectively.
- For Figure 2c, why S morphotype infected ZF does not express Tomato fluorescence? Does M. ab-S was cleared already?
Except for these questions, I am satisfying all results described.
